# Constrained Graph Variational Autoencoders for Molecule Design

**Qi Liu**[*1], **Miltiadis Allamanis**[2], **Marc Brockschmidt**[2], and **Alexander L. Gaunt**[2]

[1]Singapore University of Technology and Design
[2]Microsoft Research, Cambridge
qiliu@u.nus.edu, {miallama, mabrocks, algaunt}@microsoft.com

## Abstract

Graphs are ubiquitous data structures for representing interactions between entities. With an emphasis on applications in chemistry, we explore the task of learning to generate graphs that conform to a distribution observed in training data. We propose a variational autoencoder model in which both encoder and decoder are graph-structured. Our decoder assumes a sequential ordering of graph extension steps and we discuss and analyze design choices that mitigate the potential downsides of this linearization. Experiments compare our approach with a wide range of baselines on the molecule generation task and show that our method is successful at matching the statistics of the original dataset on semantically important metrics. Furthermore, we show that by using appropriate shaping of the latent space, our model allows us to design molecules that are (locally) optimal in desired properties.

## 1 Introduction

Structured objects such as program source code, physical systems, chemical molecules and even 3D scenes are often well represented using graphs [2, 6, 16, 25]. Recently, considerable progress has been made on building *discriminative* deep learning models that ingest graphs as inputs [4, 9, 17, 21]. Deep learning approaches have also been suggested for graph *generation*. More specifically, generating and optimizing chemical molecules has been identified as an important real-world application for this set of techniques [8, 23, 24, 28, 29].

In this paper, we propose a novel probabilistic model for graph generation that builds gated graph neural networks (GGNNs) [21] into the encoder and decoder of a variational autoencoder (VAE) [15]. Furthermore, we demonstrate how to incorporate hard domain-specific constraints into our model to adapt it for the molecule generation task. With these constraints in place, we refer to our model as a constrained graph variational autoencoder (CGVAE). Additionally, we shape the latent space of the VAE to allow optimization of numerical properties of the resulting molecules. Our experiments are performed with real-world datasets of molecules with pharmaceutical and photo-voltaic applications. By generating novel molecules from these datasets, we demonstrate the benefits of our architectural choices. In particular, we observe that (1) the GGNN architecture is beneficial for state-of-the-art generation of molecules matching chemically relevant statistics of the training distribution, and (2) the semantically meaningful latent space arising from the VAE allows continuous optimization of molecule properties [8].

The key challenge in generating graphs is that sampling directly from a joint distribution over all configurations of labeled nodes and edges is intractable for reasonably sized graphs. Therefore, a generative model must decompose this joint in some way. A straightforward approximation is to ignore correlations and model the existence and label of each edge with *independent* random

---

[*]work performed during an internship with Microsoft Research, Cambridge.

variables [5, 30, 31]. An alternative approach is to factor the distribution into a *sequence* of discrete decisions in a graph construction trace [22, 35]. Since correlations between edges are usually crucial in real applications, we pick the latter, sequential, approach in this work. Note that for molecule design, some correlations take the form of known hard rules governing molecule stability, and we explicitly enforce these rules wherever possible using a technique that masks out choices leading to illegal graphs [18, 28]. The remaining "soft" correlations (e.g. disfavoring of small cycles) are learned by our graph structured VAE.

By opting to generate graphs sequentially, we lose permutation symmetry and have to train using arbitrary graph linearizations. For computational reasons, we cannot consider all possible linearizations for each graph, so it is challenging to marginalize out the construction trace when computing the log-likelihood of a graph in the VAE objective. We design a generative model where the learned component is conditioned only on the current state of generation and not on the arbitrarily chosen path to that state. We argue that this property is intuitively desirable and show how to derive a bound for the desired log-likelihood under this model. Furthermore, this property makes the model relatively shallow and it is easy scale and train.

## 2 Related Work

Generating graphs has a long history in research, and we consider three classes of related work: Works that ignore correlations between edges, works that generate graphs sequentially and works that emphasize the application to molecule design.

**Uncorrelated generation** The Erdős-Rényi $G(n, p)$ random graph model [5] is the simplest example of this class of algorithms, where each edge exists with independent probability $p$. Stochastic block models [31] add community structure to the Erdős-Rényi model, but retain uncorrelated edge sampling. Other traditional random graph models such as those of Albert and Barabási [1], Leskovec et al. [20] do account for edge correlations, but they are hand-crafted into the models. A more modern *learned* approach in this class is GraphVAEs [30], where the decoder emits independent probabilities governing edge and node existence and labels.

**Sequential generation** Johnson [14] sidesteps the issue of permutation symmetry by considering the task of generating a graph from an auxiliary stream of information that imposes an order on construction steps. This work outlined many ingredients for the general sequential graph generation task: using GGNNs to embed the current state of generation and multi-layer perceptrons (MLPs) to drive decisions based on this embedding. Li et al. [22] uses these ingredients to build an autoregressive model for graphs without the auxiliary stream. Their model gives good results, but each decision is conditioned on a full history of the generation sequence, and the authors remark on stability and scalability problems arising from the optimization of very deep neural networks. In addition, they describe some evidence for overfitting to the chosen linearization scheme due to the strong history dependence. Our approach also uses the ingredients from Johnson [14], but avoids the training and overfitting problems using a model that is conditioned only on the current partial graph rather than on full generation traces. In addition, we combine Johnson's ingredients with a VAE that produces a meaningful latent space to enable continuous graph optimization [8].

An alternative sequential generation algorithm based on RNNs is presented in You et al. [35]. The authors point out that a dense implementation of a GGNN requires a large number $\mathcal{O}(eN^2)$ of operations to construct a graph with $e$ edges and $N$ nodes. We note that this scaling problem can be mitigated using a sparse GGNN implementation [2], which reduces complexity to $\mathcal{O}(e^2)$.

**Molecule design** Traditional *in silico* molecule design approaches rely on considerable domain knowledge, physical simulation and heuristic search algorithms (for a recent example, see Gómez-Bombarelli et al. [7]). Several deep learning approaches have also been tailored to molecule design, for example [13] is a very promising method that uses a library of frequent (ring-containing) fragments to reduce the graph generation process to a tree generation process where nodes represent entire fragments. Alternatively, many methods rely on the SMILES linearization of molecules [33] and use RNNs to generate new SMILES strings [8, 23, 24, 29]. A particular challenge of this approach is to ensure that the generated strings are syntactically valid under the SMILES grammar. The Grammar VAE uses a mask to impose these constraints during generation and a similar technique is applied

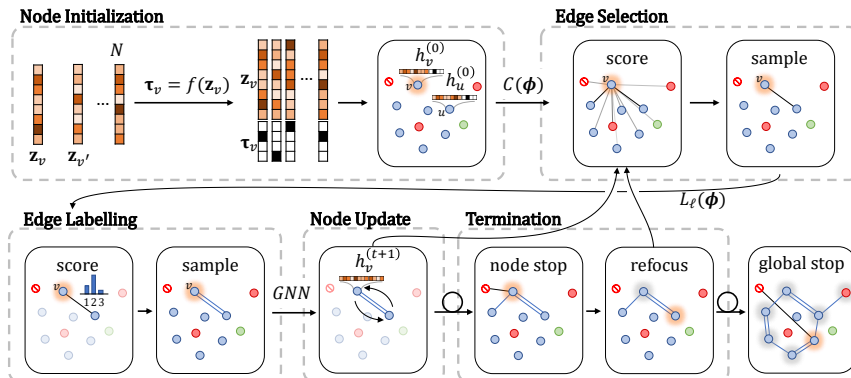

Figure 1: Illustration of the phases of the generative procedure. Nodes are initialized with latent variables and then we enter a loop between edge selection, edge labelling and node update steps until the special stop node $\oslash$ is selected. We then refocus to a new node or terminate if there are no candidate focus nodes in the connected component. A looped arrow indicates that several loop iterations may happen between the illustrated steps.

for general graph construction in Samanta et al. [28]. Our model also employs masking that, among other things, ensures that the molecules we generate can be converted to syntactically valid SMILES strings.

## 3   Generative Model

Our generative procedure is illustrated in Fig. 1. The process is seeded with $N$ vectors $\mathbf{z}_v$ that together form a latent "specification" for the graph to be generated ($N$ is an upper bound on the number of nodes in the final graph). Generation of edges between these nodes then proceeds using two decision functions: `focus` and `expand`. In each step the `focus` function chooses a focus node to visit, and then the `expand` function chooses edges to add from the focus node. As in breadth-first traversal, we implement `focus` as a deterministic queue (with a random choice for the initial node).

Our task is thus reduced to learning the `expand` function that enumerates new edges connected to the currently focused node. One design choice is to make `expand` condition upon the full history of the generation. However, this has both theoretical and practical downsides. Theoretically, this means that the learned model is likely to learn to reproduce generation traces. This is undesirable, since the underlying data usually only contains fully formed graphs; thus the exact form of the trace is an artifact of the implemented data preprocessing. Practically, this would lead to extremely deep computation graphs, as even small graphs easily have many dozens of edges; this makes training of the resulting models very hard as mentioned in mentioned in Li et al. [22]. Hence, we condition `expand` only upon the partial graph structure $\mathcal{G}^{(t)}$ generated so far; intuitively, this corresponds to learning how to complete a partial graph without using any information about how the partial graph was generated. We now present the details of each stage of this generative procedure.

**Node Initialization**   We associate a state $\mathbf{h}_v^{(t=0)}$ with each node $v$ in a set of initially unconnected nodes. Specifically, $\mathbf{z}_v$ is drawn from the $d$-dimensional standard normal $\mathcal{N}\left(\mathbf{0}, \mathbf{I}\right)$, and $\mathbf{h}_v^{(t=0)}$ is the concatenation $[\mathbf{z}_v, \boldsymbol{\tau}_v]$, where $\boldsymbol{\tau}_v$ is an interpretable one-hot vector indicating the node type. $\boldsymbol{\tau}_v$ is derived from $\mathbf{z}_v$ by sampling from the softmax output of a learned mapping $\boldsymbol{\tau}_v \sim f(\mathbf{z}_v)$ where $f$ is a neural network[2]. The interpretable component of $\mathbf{h}_v^{(t=0)}$ gives us a means to enforce hard constraints during generation.

From these node-level variables, we can calculate global representations $\mathbf{H}^{(t)}$ (the average representation of nodes in the connected component at generation step $t$), and $\mathbf{H}_{\text{init}}$ (the average representation of all nodes at $t=0$). In addition to $N$ working nodes, we also initialize a special "stop node" to a learned representation $\mathbf{h}_\oslash$ for managing algorithm termination (see below).

**Node Update** Whenever we obtain a new graph $\mathcal{G}^{(t+1)}$, we discard $\mathbf{h}_v^{(t)}$ and compute new representations $\mathbf{h}_v^{(t+1)}$ for all nodes taking their (possibly changed) neighborhood into account. This is implemented using a standard gated graph neural network (GGNN) $G_{\text{dec}}$ for $S$ steps[3], which is defined as a recurrent operation over messages $\mathbf{m}_v^{(s)}$.

$$\mathbf{m}_v^{(0)} = \mathbf{h}_v^{(0)} \qquad \mathbf{m}_v^{(s+1)} = \text{GRU}\left[\mathbf{m}_v^{(s)}, \sum_{v \overset{\ell}{\leftrightarrow} u} E_\ell(\mathbf{m}_u^{(s)})\right] \qquad \mathbf{h}_v^{(t+1)} = \mathbf{m}_v^{(S)}$$

Here the sum runs over all edges in the current graph and $E_\ell$ is an edge-type specific neural network[4] We also augment our model with a master node as described by Gilmer et al. [6]. Note that since $\mathbf{h}_v^{(t+1)}$ is computed from $\mathbf{h}_v^{(0)}$ rather than $\mathbf{h}_v^{(t)}$, the representation $\mathbf{h}_v^{(t+1)}$ is independent of the generation history of $\mathcal{G}^{(t+1)}$.

**Edge Selection and Labelling** We first pick a focus node $v$ from our queue. The function `expand` then selects edges $v \overset{\ell}{\nleftrightarrow} u$ from $v$ to $u$ with label $\ell$ as follows. For each non-focus node $u$, we construct a feature vector $\boldsymbol{\phi}_{v,u}^{(t)} = [\mathbf{h}_v^{(t)}, \mathbf{h}_u^{(t)}, d_{v,u}, \mathbf{H}_{\text{init}}, \mathbf{H}^{(t)}]$, where $d_{v,u}$ is the graph distance between $v$ and $u$. This provides the model with both local information for the focus node $v$ and the candidate edge $(\mathbf{h}_v^{(t)}, \mathbf{h}_u^{(t)})$, and global information regarding the original graph specification ($\mathbf{H}_{\text{init}}$) and the current graph state ($\mathbf{H}^{(t)}$). We use these representations to produce a distribution over candidate edges:

$$p(v \overset{\ell}{\nleftrightarrow} u \mid \boldsymbol{\phi}_{v,u}^{(t)}) = p(\ell \mid \boldsymbol{\phi}_{v,u}^{(t)}, v \leftrightarrow u) \cdot p(v \leftrightarrow u \mid \boldsymbol{\phi}_{v,u}^{(t)}).$$

The factors are calculated as softmax outputs from neural networks $C$ (determining the target node for an edge) and $L_\ell$ (determining the type of the edge):[5]

$$p(v \leftrightarrow u \mid \boldsymbol{\phi}_{v,u}^{(t)}) = \frac{M_{v \leftrightarrow u}^{(t)} \exp[C(\boldsymbol{\phi}_{v,u}^{(t)})]}{\sum_w M_{v \leftrightarrow w}^{(t)} \exp[C(\boldsymbol{\phi}_{v,w}^{(t)})]}, \quad p(\ell \mid \boldsymbol{\phi}_{v,u}^{(t)}) = \frac{m_{v \overset{\ell}{\nleftrightarrow} u}^{(t)} \exp[L_\ell(\boldsymbol{\phi}_{v,u}^{(t)})]}{\sum_k m_{v \overset{k}{\nleftrightarrow} u}^{(t)} \exp[L_k(\boldsymbol{\phi}_{v,u}^{(t)})]}. \quad (1)$$

$M_{v \leftrightarrow u}^{(t)}$ and $m_{v \overset{\ell}{\nleftrightarrow} u}^{(t)}$ are binary masks that forbid edges that violate constraints. We discuss the construction of these masks for the molecule generation case in Sect. 5.2. New edges are sampled from these distributions, and any nodes that are connected to the graph for the first time are added to the `focus` queue. Note that we only consider undirected edges in this paper, but it is easy to extend the model to directed graphs.

**Termination** We keep adding edges to a node $v$ using `expand` and $G_{\text{dec}}$ until an edge to the stop node is selected. Node $v$ then loses focus and becomes "closed" (mask $M$ ensures that no further edges will ever be made to $v$). The next focus node is selected from the `focus` queue. In this way, a single connected component is grown in a breadth-first manner. Edge generation continues until the queue is empty (note that this may leave some unconnected nodes that will be discarded).

## 4 Training the Generative Model

The model from Sect. 3 relies on a latent space with semantically meaningful points concentrated in the region weighted under the standard normal, and trained networks $f$, $C$, $L_\ell$ and $G_{\text{dec}}$. We train these in a VAE architecture on a large dataset $\mathcal{D}$ of graphs. Details of this VAE are provided below.

### 4.1 Encoder

The encoder of our VAE is a GGNN $G_{\text{enc}}$ that embeds each node in an input graph $\mathcal{G}$ to a diagonal normal distribution in $d$-dimensional latent space parametrized by mean $\boldsymbol{\mu}_v$ and standard deviation $\boldsymbol{\sigma}_v$ vectors. The latent vectors $\mathbf{z}_v$ are sampled from these distributions, and we construct the usual VAE regularizer term measuring the KL divergence between the encoder distribution and the standard Gaussian prior: $\mathcal{L}_{\text{latent}} = \sum_{v \in \mathcal{G}} \text{KL}(\mathcal{N}(\boldsymbol{\mu}_v, \text{diag}(\boldsymbol{\sigma}_v)^2) \| \mathcal{N}(\mathbf{0}, \mathbf{I}))$.

## 4.2 Decoder

The decoder is the generative procedure described in Sect. 3, and we condition generation on a latent sample from the encoder distribution during training. We supervise training of the overall model using generation traces extracted from graphs in $\mathcal{D}$.

**Node Initialization**   To obtain initial node states $\mathbf{h}_v^{(t=0)}$, we first sample a node specification $\mathbf{z}_v$ for each node $v$ and then independently for each node we generate the label $\boldsymbol{\tau}_v$ using the learned function $f$. The probability of re-generating the labels $\boldsymbol{\tau}_v^*$ observed in the encoded graph is given by a sum over node permutations $\mathcal{P}$:

$$p(\mathcal{G}^{(0)} \mid \mathbf{z}) = \sum_{\mathcal{P}} p(\boldsymbol{\tau} = \mathcal{P}(\boldsymbol{\tau^*}) \mid \mathbf{z}) > \prod_v p(\boldsymbol{\tau}_v = \boldsymbol{\tau}_v^* \mid \mathbf{z}_v).$$

This inequality provides a lower bound given by the single contribution from the ordering used in the encoder (recall that in the encoder we know the node type $\boldsymbol{\tau}_v^*$ from which $\mathbf{z}_v$ was generated). A set2set model [32] could improve this bound.

**Edge Selection and Labelling**   During training, we provide supervision on the sequence of edge additions based on breadth-first traversals of each graph in the dataset $\mathcal{D}$. Formally, to learn a distribution over *graphs* (and not graph generation traces), we would need to train with an objective that computes the log-likelihood of each graph by marginalizing over all possible breadth-first traces. This is computationally intractable, so in practice we only compute a Monte-Carlo estimate of the marginal on a small set of sampled traces. However, recall from Sect. 3 that our `expand` model is not conditioned on full traces, and instead only considers the partial graph generated so far. Below we outline how this intuitive design formally affects the VAE training objective.

Given the initial collection of unconnected nodes, $\mathcal{G}^{(0)}$, from the initialization above, we first use Jensen's inequality to show that the log-likelihood of a graph $\mathcal{G}$ is loosely lower bounded by the expected log-likelihood of all the traces $\Pi$ that generate it.

$$\log p(\mathcal{G} \mid \mathcal{G}^{(0)}) = \log \sum_{\pi \in \Pi} p(\pi \mid \mathcal{G}^{(0)}) \geq \log(|\Pi|) + \frac{1}{|\Pi|} \sum_{\pi \in \Pi} \log p(\pi \mid \mathcal{G}^{(0)}) \qquad (2)$$

We can decompose each full generation trace $\pi \in \Pi$ into a sequence of steps of the form $(t, v, \epsilon)$, where $v$ is the current focus node and $\epsilon = v \not\leftrightarrow u$ is the edge added at step $t$:

$$\log p(\pi \mid \mathcal{G}^{(0)}) = \sum_{(t,v,\epsilon) \in \pi} \left\{ \log p(v \mid \pi, t) + \log p(\epsilon \mid \mathcal{G}^{(t-1)}, v) \right\}$$

The first term corresponds to the choice of $v$ as focus node at step $t$ of trace $\pi$. As our `focus` function is fixed, this choice is uniform in the first focus node and then deterministically follows a breadth-first queuing system. A summation over this term thus evaluates to the constant $\log(1/N)$.

As discussed above, the second term is only conditioned on the current graph (and not the whole generation history $\mathcal{G}^{(0)} \ldots \mathcal{G}^{(t-1)}$). To evaluate it further, we consider the set of generation states $\mathcal{S}$ of all valid state pairs $s = (\mathcal{G}^{(t)}, v)$ of a partial graph $\mathcal{G}^{(t)}$ and a focus node $v$. We use $|s|$ to denote the multiplicity of state $s$ in $\Pi$, i.e., the number of traces that contain graph $\mathcal{G}^{(t)}$ and focus on node $v$. Let $\mathcal{E}_s$ denote all edges that could be generated at state $s$, i.e., the edges from the focus node $v$ that are present in the graph $\mathcal{G}$ from the dataset, but are not yet present in $\mathcal{G}^{(t)}$. Then, each of these appears uniformly as the next edge to generate in a trace for all $|s|$ occurrences of $s$ in a trace from $\Pi$,

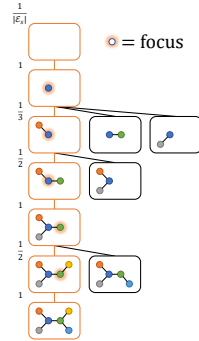

and therefore, we can rearrange a sum over paths into a sum over steps:

$$\frac{1}{|\Pi|} \sum_{\pi \in \Pi} \sum_{(t,v,\epsilon) \in \pi} \log p(\epsilon \mid s) = \frac{1}{|\Pi|} \sum_{s \in \mathcal{S}} \sum_{\epsilon \in \mathcal{E}_s} \frac{|s|}{|\mathcal{E}_s|} \log p(\epsilon \mid s)$$

$$= \mathbb{E}_{s \sim \Pi} \left[ \frac{1}{|\mathcal{E}_s|} \sum_{\epsilon \in \mathcal{E}_s} \log p(\epsilon \mid s) \right]$$

Figure 2: Steps considered in our model.

Here we use that $|s|/|\Pi|$ is the probability of observing state $s$ in a random draw from all states in $\Pi$. We use this expression in Eq. 2 and train our VAE with a reconstruction loss $\mathcal{L}_{\text{recon.}} = \sum_{\mathcal{G} \in \mathcal{D}} \log \left[ p(\mathcal{G} \mid \mathcal{G}^{(0)}) \cdot p(\mathcal{G}^{(0)} \mid \mathbf{z}) \right]$ ignoring additive constants.

We evaluate the expectation over states $s$ using a Monte Carlo estimate from a set of enumerated generation traces. In practice, this set of paths is very small (e.g. a single trace) resulting in a high variance estimate. Intuitively, Fig. 2 shows that rather than requiring the model to exactly reproduce each step of the sampled paths (orange) our objective does not penalize the model for choosing any valid expansion at each step (black).

## 4.3 Optimizing Graph Properties

So far, we have described a generative model for graphs. In addition, we may wish to perform (local) optimization of these graphs with respect to some numerical property, $Q$. This is achieved by gradient ascent in the continuous latent space using a differentiable gated regression model

$$R(\mathbf{z}_v) = \sum_v \sigma(g_1(\mathbf{z}_v)) \cdot g_2(\mathbf{z}_v),$$

where $g_1$ and $g_2$ are neural networks[6] and $\sigma$ is the sigmoid function. Note that the combination of $R$ with $G_{\text{enc}}$ (i.e., $R(G_{\text{enc}}(\mathcal{G}))$) is exactly the GGNN regression model from Gilmer et al. [6]. During training, we use an $L_2$ distance loss $\mathcal{L}_Q$ between $R(\mathbf{z}_v)$ and the labeled properties $Q$. This regression objective shapes the latent space, allowing us to optimize for the property $Q$ in it. Thus, at test time, we can sample an initial latent point $\mathbf{z}_v$ and then use gradient ascent to a locally optimal point $\mathbf{z}_v^*$ subject to an $L_2$ penalty that keeps the $\mathbf{z}_v^*$ within the standard normal prior of the VAE. Decoding from the point $\mathbf{z}_v^*$ then produces graphs with an optimized property $Q$. We show this in our experiments in Sect. 6.2.

## 4.4 Training objective

The overall objective is $\mathcal{L} = \mathcal{L}_{\text{recon.}} + \lambda_1 \mathcal{L}_{\text{latent}} + \lambda_2 \mathcal{L}_Q$, consisting of the usual VAE objective (reconstruction terms and regularization on the latent variables) and the regression loss. Note that we allow deviation from the pure VAE loss ($\lambda_1 = 1$) following Yeung et al. [34].

# 5 Application: Molecule Generation

In this section, we describe additional specialization of our model for the application of generating chemical molecules. Specifically, we outline details of the molecular datasets that we use and the domain specific masking factors that appear in Eq. 1.

## 5.1 Datasets

We consider three datasets commonly used in the evaluation of computational chemistry approaches:

- QM9 [26, 27], an enumeration of $\sim 134$k stable organic molecules with up to 9 heavy atoms (carbon, oxygen, nitrogen and fluorine). As no filtering is applied, the molecules in this dataset only reflect basic structural constraints.
- ZINC dataset [12], a curated set of 250k commercially available drug-like chemical compounds. On average, these molecules are bigger ($\sim 23$ heavy atoms) and structurally more complex than the molecules in QM9.
- CEPDB [10, 11], a dataset of organic molecules with an emphasis on photo-voltaic applications. The contained molecules have $\sim 28$ heavy atoms on average and contain six to seven rings each. We use a subset of the full database containing 250k randomly sampled molecules.

For all datasets we kekulize the molecules so that the only edge types to consider are single, double and triple covalent bonds and we remove all hydrogen atoms. In the encoder, molecular graphs are presented with nodes annotated with onehot vectors $\boldsymbol{\tau}_v^*$ indicating their atom type and charge.

(a)

| | Measure | 2: CGVAE | 3: [22] | 4: LSTM | 5: [8] | 6: [18] | 7: [30] | 8: [28] |
|---|---|---|---|---|---|---|---|---|
| QM9 | % valid | 100 | - | 94.78 | 10.00 | 30.00 | 61.00 | 98.00 |
| | % novel | 94.35 | - | 82.98 | 90.00 | 95.44 | 85.00 | 100 |
| | % unique | 98.57 | - | 96.94 | 67.50 | 9.30 | 40.90 | 99.86 |
| ZINC | % valid | 100 | 89.20 | 96.80 | 17.00 | 31.00 | 14.00 | - |
| | % novel | 100 | 89.10 | 100 | 98.00 | 100 | 100 | - |
| | % unique | 99.82 | 99.41 | 99.97 | 30.98 | 10.76 | 31.60 | - |
| CEPDB | % valid | 100 | - | 99.61 | 8.30 | 0.00 | - | - |
| | % novel | 100 | - | 92.43 | 90.05 | - | - | - |
| | % unique | 99.62 | - | 99.56 | 80.99 | - | - | - |

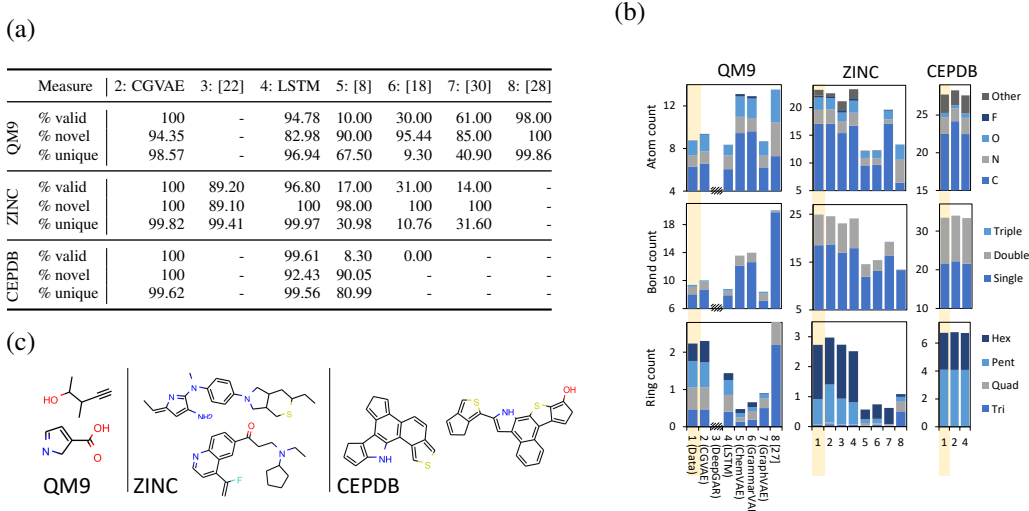

(b)

(c)

QM9   ZINC   CEPDB

Figure 3: Overview of statistics of sampled molecules from a range of generative models trained on different datasets. In (b) We highlight the target statistics of the datasets in yellow and use the numbers 2, ..., 7 to denote different models as shown in the axis key. A hatched box indicates where other works do not supply benchmark results. Two samples from our model on each dataset are shown in (c), with more random samples given in supplementary material A.

## 5.2 Valency masking

Valency rules impose a strong constraint on constructing syntactically valid molecules[7]. The valency of an atom indicates the number of bonds that that atom can make in a stable molecule, where edge types "double" and "triple" count for 2 and 3 bonds respectively. In our data, each node type has a fixed valency given by known chemical properties, for example node type "O" (an oxygen atom) has a valency of 2 and node type "$O^-$" (an oxygen ion) has valency of 1. Throughout the generation process, we use masks $M$ and $m$ to guarantee that the number of bonds $b_v$ at each node never exceeds the valency $b_v^*$ of the node. If $b_v < b_v^*$ at the end of generation we link $b_v^* - b_v$ hydrogen atoms to node $v$. In this way, our generation process always produces syntactically valid molecules (we define syntactic validity as the ability to parse the graph to a SMILES string using the RDKit parser [19]). More specifically, $M_{v \leftrightarrow u}^{(t)}$ also handles avoidance of edge duplication and self loops, and is defined as:

$$M_{v \leftrightarrow u}^{(t)} = \mathbb{1}(b_v < b_v^*) \times \mathbb{1}(b_u < b_u^*) \times \mathbb{1}(\text{no } v \leftrightarrow u \text{ exists}) \times \mathbb{1}(v \neq u) \times \mathbb{1}(u \text{ is not closed}), \quad (3)$$

where $\mathbb{1}$ is an indicator function, and as a special case, connections to the stop node are always unmasked. Further, when selecting the label for a chosen edge, we must again avoid violating the valency constraint, so we define $m_{v \not\leftrightarrow u}^{(t)} = M_{v \leftrightarrow u}^{(t)} \times \mathbb{1}(b_u^* - b_u \leq \ell)$, using $\ell = 1, 2, 3$ to indicate single, double and triple bond types respectively

## 6 Experiments

We evaluate baseline models, our model (CGVAE) and a number of ablations on the two tasks of molecule generation and optimization[8].

## 6.1 Generating molecules

As baseline models, we consider the deep autoregressive graph model (that we refer to as DeepGAR) from [22], a SMILES generating LSTM language model with 256 hidden units (reduced to 64 units

for the smaller QM9 dataset), ChemVAE [8], GrammarVAE [18], GraphVAE [30], and the graph model from [28]. We train these and on our three datasets and then sample 20k molecules from the trained models (in the case of [22, 28], we obtained sets of sampled molecules from the authors).

We analyze the methods using two sets of metrics. First in Fig. 3(a) we show metrics from existing work: syntactic validity, novelty (i.e. fraction of sampled molecules not appearing in the training data) and uniqueness (i.e. ratio of sample set size before and after deduplication of identical molecules). Second, in Fig. 3(b) we introduce new metrics to measure how well each model captures the distribution of molecules in the training set. Specifically, we measure the average number of each atom type and each bond type in the sampled molecules, and we count the average number of 3-, 4-, 5-, and 6-membered cycles in each molecule. This latter metric is chemically relevant because 3- and 4-membered rings are typically unstable due to their high ring strain. Fig. 3(c) shows 2 samples from our model for each dataset and we show more samples of generated molecules in the supplementary material.

The results in Fig. 3 show that CGVAE is excellent at matching graph statistics, while generating valid, novel and unique molecules for all datasets considered (additional details are found in supplementary material B and C). The only competitive baselines are DeepGAR from Li et al. [22] and an LSTM language model. Our approach has three advantages over these baselines: First, whereas >10% of ZINC-like molecules generated by DeepGAR are invalid, our masking mechanism guarantees molecule validity. An LSTM is surprisingly effective at generating valid molecules, however, LSTMs do not permit the injection of domain knowledge (e.g. valence rules or requirement for the existance of a particular scaffold) because meaningful constraints cannot be imposed on the flat SMILES representation during generation. Second, we train a shallow model on breadth-first steps rather than full paths and therefore do not experience problems with training instability or overfitting that are described in Li et al. [22]. Empirical indication for overfitting in DeepGAR is seen by the fact that Li et al. [22] achieves the lowest novelty score on the ZINC dataset, suggesting that it more often replays memorized construction traces. It is also observed in the LSTM case, where on average 60% of each generated SMILES string is copied from the nearest neighbour in the training set. Converting our generated graphs to SMILES strings reveals only 40% similarity to the nearest neighbour in the same metric. Third we are able to use our continuous latent space for molecule optimization (see below).

We also perform an ablation study on our method. For brevity we only report results using our ring count metrics, and other statistics show similar behavior. From all our experiments we highlight three aspects that are important choices to obtain good results, and we report these in ablation experiments A, B and C in Fig. 4. In experiment A we remove the distance feature $d_{v,u}$ from $\phi$ and see that this harms performance on the larger molecules in the ZINC dataset. More interestingly, we see poor results in experiment B where we make an independence assumption on edge generation (i.e. use features $\phi$ to calculate independent probabilities for all possible edges and sample an entire molecule in one step). We also see poor results in experiment C where we remove the GGNN from the decoder (i.e. perform sequential construction with $\mathbf{h}_v^{(t)} = \mathbf{h}_v^{(0)}$).

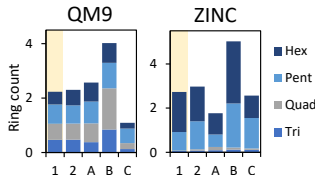

Figure 4: Ablation study using the ring metric. 1 indicates statistics of the datasets, 2 of our model and A,B,C of the ablations discussed in the text.

This indicates that the choice to perform sequential decoding with GGNN node updates before each decision are the keys to the success of our model.

## 6.2 Directed molecule generation

Finally, we show that we can use the VAE structure of our method to direct the molecule generation towards especially interesting molecules. As discussed in Sect. 4.3 (and first shown by Gómez-Bombarelli et al. [8] in this setting), we extend our architecture to predict the *Quantitative Estimate of Drug-Likeness* (QED) directly from latent space. This allows us to generate molecules with very high QED values by performing gradient ascent in the latent space using the trained QED-scoring network. Fig. 5 shows an interpolation sequence from a point in latent space with an low QED value (which ranges between 0 and 1) to the local maximum. For each point in the sequence, the figure shows a generated molecule, the QED value our architecture predicts for this molecule, as well as the QED value computed by RDKit.

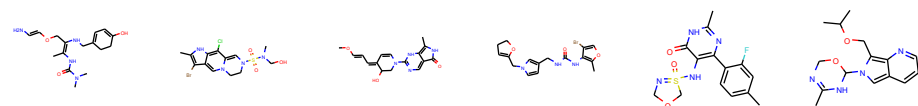

| | | | | | | |
|---|---|---|---|---|---|---|
| Pred. QED | 0.5686 | 0.6685 | 0.7539 | 0.8376 | 0.9013 | 0.9271 |
| Real QED | 0.5345 | 0.6584 | 0.7423 | 0.8298 | 0.8936 | 0.9383 |

Figure 5: Trajectory of QED-directed optimization in latent space. Additional examples are shown in supplementary material D.

## 7 Conclusion

We proposed CGVAE, a sequential generative model for graphs built from a VAE with GGNNs in the encoder and decoder. Using masks that enforce chemical rules, we specialized our model to the application of molecule generation and achieved state-of-the-art generation and optimization results. We introduced basic statistics to validate the quality of our generated molecules. Future work will need to link to the chemistry community to define additional metrics that further guide the construction of models and datasets for real-world molecule design tasks.

## Footnotes

[2]We implement $f$ as a linear classifier from the 100 dimensional latent space to one of the node type classes.

[3]Our experiments use $S = 7$.

[4]In our implementation, $E_\ell$ is a dimension-preserving linear transformation.

[5]$C$ and $L_\ell$ are fully connected networks with a single hidden layer of 200 units and ReLU non-linearities.

[6]In our experiments, both $g_1$ and $g_2$ are implemented as linear transformations that project to scalars.

[7]Note that more complex domain knowledge e.g. Bredt's rule [3] could also be handled in our model but we do not implement this here.

[8]Our implementation of CGVAE can be found at `https://github.com/Microsoft/constrained-graph-variational-autoencoder`.

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
