[Supplementary Material · CGVAE_supplementary.pdf]

# Supplementary Material: CGVAE for Molecule Design

## A Molecule Samples

We provide 25 random samples from our model for qualitative comparison with samples from each training dataset.

# B    Effect of multiple training paths

Section 4.2 describes how we should enumerate all breadth first graph generation traces, break these traces into state transitions and then randomly sample state transitions to give the Monte Carlo estimate of the reconstruction loss. However, for computational efficiency, in the presented experiments we provide only a single trace containing $E$ transitions (where $E$ is the number of edges in the final molecule including edges to the stop node). Figure 6 shows an additional experiment (CGVAE (50)) where we enumerate 50 traces for each molecule and sample $E$ transitions from this enumeration (so the final dataset size is the same). While increasing the number of traces considered produces a small improvement in the matching of ring statistics in the sampled molecules, it is not clear that this benefit is worth the considerable computational overhead required in preparing the dataset.

Figure 6: Distribution of structural metrics for models trained on ZINC. We represent the distribution of each property over samples as a point at the mean with error bars covering 1 standard deviation. We calculate the metrics on the raw dataset (ZINC) and samples drawn from CGVAE trained on 1 (CGVAE (1)) or 50 (CGVAE (50)) generation traces.

# C  Additional Molecular Properties

Here we provide histograms of the following molecular properites of the sampled molecules for our method and the DeepGAR and LSTM baselines:

| Property | RDKit Implementation | Description |
|---|---|---|
| Molecular Weight | Chem.Descriptors.MolWt | The isotope-averaged molecular weight in atomic mass units. |
| Bertz Complexity | Chem.GraphDescriptors.BertzCT | A topological index meant to quantify complexity of molecules. |
| H donor count | Chem.Lipinski.NumHDonors | Number of heavy atoms bonded to H atoms that can form Hydrogen bonds. |
| H acceptor count | Chem.Lipinski.NumHAcceptors | Number of heavy atoms with lone electron pairs that can form Hydrogen bonds. |
| Rotatable bond count | Chem.Lipinski.NumRotatableBonds | Rotatable bonds are any single bond, not in a ring, bound to a nonterminal heavy atom. |
| Partition coefficient | Chem.Crippen.MolLogP | The octanol/water logP partition coefficient according to Wildman and Crippen 1999. |
| Topological polar surface area | Chem.rdMolDescriptors.CalcTPSA | The total exposed surface area of polar atoms in a molecule including attached Hydrogens (in square angstroms). |

# D  Optimization trajectories

We provide additional QED optimization trajectories for our model trained on the ZINC dataset.

| | | | | | | |
|---|---|---|---|---|---|---|
| Pred. QED | 0.5073 | 0.6249 | 0.6683 | 0.7375 | 0.8367 | 0.8927 |
| Real QED | 0.4419 | 0.5973 | 0.6789 | 0.7497 | 0.8186 | 0.8796 |

| | | | | | | |
|---|---|---|---|---|---|---|
| Pred. QED | 0.4368 | 0.5046 | 0.7017 | 0.7683 | 0.8906 | 0.9427 |
| Real QED | 0.4188 | 0.5131 | 0.6910 | 0.7889 | 0.8709 | 0.9309 |