[Reviews · NeurIPS 2018]

Reviewer 1



Summary: This paper describes a model for generating graph-structured data, with molecule generation being the example task. This model is based around a variational autoencoder whose encoder/decoder are designed to handle graph-structured data. The decoder builds a graph sequentially by starting from an arbitrary node and sampling edges to other nodes, which are placed in a queue; upon sampling an edge to a “stopping node,” the next node is taken from the queue and the process continues until there are no more nodes to expand. The distributions from which these samples are taken are a function of the graph state (notably, not the specific steps taken to arrive at the current state), where the state vectors are encoded using a gated graph neural network (GGNN). Additionally, masking functions can be specified that serve as hard constraints on the sorts of edges that may be sampled (in case these would lead to graphs that are disallowed, e.g. that would lead to impossible molecules). The encoder is also comprised of a GGNN. Training is accomplished using a standard VAE objective; however, for computational efficiency reasons the reconstruction loss is only estimated. Additionally, a term can be added to the loss which is a function of the latent representation that attempts to optimize some numerical quantity related to the graphs being generated; at test time, gradient ascent can be used to find a latent representation with a “better” value of this quantity. Experiments were run on three molecule datasets, with comparisons being made against a variety of graph generation baselines regarding generated molecule uniqueness, validity, and novelty, as well as how the distributions of generated molecule statistics compare against the training dataset. Comments: (using the same citation numbering as in the paper) This work is primarily a synthesis of ideas and techniques explored in other works: using a GGNN to model graphs as in [21]; training using a VAE objective as in [27]; enforcing hard constraints on generated graphs as in [27]; and learning a model that is invariant to graph construction order as in [27]. The real strength of this paper is that it combines these ideas and makes tweaks that allow the model to function well in practice, which seems to be primarily a combination of the modeling strength of GGNN (demonstrated by comparing the GGNN-based models against the non GGNN-based models during experiments) with the masking of invalid edges during generation and construction-order invariance. The provided experiments provide sufficient evidence that the proposed model has absorbed the benefits of these prior models (though it would have been nice to see more comparisons against [21] to ensure that the better results of this model are not an artifact of the one dataset used). There are a few things I felt were missing that would have strengthened this paper a bit. The optimization of task-specific metrics (as described in section 4.3) was somewhat underexplored - section 6.2 presents a small experiment demonstrating that it is possible to learn a network that can optimize for a given metric, but only a few samples are provided. A full quantitative evaluation of this portion of the model would have been interesting to see (e.g. an ablation comparing the average metric for generated molecules with and without this additional modeling step and whether doing so hurts performance on other metrics at all). Additionally, details of training were completely omitted, including training hyperparameters and how they were chosen - this hurts reproducibility. Some further ablations related to training details, e.g. whether or not the number of path samples taken of the training data, would also have been interesting to see. Overall, though, these omissions do not detract significantly, and I feel that this paper is very worthy of submission. It should also be noted that this paper is very well written, with the specific details presented in an exceptionally clear manner. None of the details presented were confusing. POST AUTHOR RESPONSE EDIT: Thanks for the response. My opinions remain more or less the same, as your response did not address my primary concerns - that a comparison against Li et al on more datasets would further justify your claims and that the experiments in section 6.2 are somewhat preliminary in nature. However, I do appreciate the additional ablation experiments you did run, and your discussion of the evaluation metrics was important and should probably be added to the discussion presented during the experiments.

Reviewer 2



Summary: The authors present a generative method for graphs tailored for molecules. Quality: Although the method is technically sound, there are many design choices that should be justified are backed more with theoretical or empirical work. The authors have knowledge of the related works and cite and mention those in the related works section. Overall, the method is very similar to that of Li et al. [1], but less rigorously evaluated and restricted to molecular graphs. Because of the lacking experiments, it is difficult to decide whether this method represents an advance in area of graph generation. Originality: The method is similar to the approach of Li et al. [1] and contains a few novel parts, for example the non-recurrent generation process. Overall, I consider this as a variation of existing algorithms. The method is also restricted to molecular graphs, which makes the work rather apt for a conference or journal in cheminformatics. Significance: The significance of this work is unclear because of the lacking experiments. The reported performance measures in “Figure 3. Table (a)” come without error bars and only report % of valid molecules, % of novel molecules and % of unique molecules. The quality of the graphs is not included in this metrics. Li et al. [1] use the negative log-likelihood as a criterion. “Figure 3. Table (b)” reports the distribution of selected structural elements of graphs, which could - via visual inspection - indicate an improvement. The authors should use a relevant scalar performance criterion (e.g. [2]) to assess and compare the methods and report the results with error bars and/or significance tests. Clarity: The paper is understandable, well structured and well written and the notation follows the standards in the community. However, in the experiments section, the writing lacks clarity (see also at “Significance”). The authors should improve the structuring of the experiments section and the presentation of the results. Minor comments: Some notation could be explained more, e.g. the double arrow symbol Notation and explanation arising from program code, such as “expand” and “focus”, should be avoided Missing related work: [3] [1] Li, Y., Vinyals, O., Dyer, C., Pascanu, R., & Battaglia, P. (2018). Learning deep generative models of graphs. arXiv preprint arXiv:1803.03324. [2] Preuer, K., Renz, P., Unterthiner, T., Hochreiter, S., & Klambauer, G. (2018). Fr\'echet ChemblNet Distance: A metric for generative models for molecules. arXiv preprint arXiv:1803.09518. [3] Popova, M., Isayev, O., & Tropsha, A. (2017). Deep reinforcement learning for de-novo drug design. arXiv preprint arXiv:1711.10907.

Reviewer 3



The authors describe a graph variational autoencoder with applications to molecule design. The proposed model specifies latent space variables that encode the nodes in the graph. The graph generation is then performed by iterating over nodes, and connecting the current node with others until a connection with an empty node is obtained. At the end, some nodes may not be connected to a bigger connected component and they are discarded. Simple rules guarantee that the proposed model can generate always valid molecules. The proposed generative model is evaluated in different tasks including the generation of synthetic molecules with similar properties as real-world ones and on molecule optimization. Quality: I found the experiments on directed molecule generation (section 6.2) quite limited. The authors only show a single example. It would be good to compare with simple baselines here and to include results for many different optimizations from different points in latent space. Clarity: The paper is clearly written and it is easy to read. Originality: Similar graph generative models have been proposed. However, this is the first one based on a variational autoencoder. The way in which they address the problem of being insensitive to the ordering in which the graph is generated (for example, Figure 2) is novel. The same happens with the masking approach to guarantee that the generated molecules are valid. The authors should include the following paper in their related work section: W. Jin, R. Barzilay, and T. Jaakkola. Junction tree variational autoencoder for molecular graph generation. In International Conference on Machine Learning (ICML), 2018. Significance: The proposed model includes significant novel contributions as mentioned above. I would only indicate the need to improve section 6.2 as mentioned above and perhaps include more experiments showing the gains of the different contributions mentioned above: masking, order independent training (figure 2).

Reviewer 4



The authors propose an interesting and reasonable variational autoencoder-based model for graph generation. The model works by sampling all potential nodes in the graph at once, which are then connected stepwise via edges in a breadth-first traversal. The model is then evaluated in several tasks measuring the properties of the generated graphs, and an optimization task. It is commendable that the authors performed ablation studies, which justify several nonobvious steps in the model design. Overall, I think this paper is a good ML paper, especially compared to other recent work on novel generative models for molecule design. Given the importance of graph structures for almost any domain, I think it would be worth presenting this work at NIPS. However, a few additional experiments should be considered to make the paper stronger, and to back up the claim of state of the art, which I currently don’t see fully supported. If the additional suggested experiments (in particular a SMILES-LSTM baseline) are provided, I would increase my evaluation from 7 to 8, regardless of the outcome of the comparison experiments. Data: The use of the QM9 dataset is rather unconvincing. Since QM9 is a small, enumerated dataset, based on hard constraints (everything which is not in this dataset is supposed to be very unstable), a proper generative model, which has learned the underlying distribution, should not generate anything outside of this data. Also, when it is practically feasible to enumerate a whole set of molecules, why is a generative model needed? Thankfully, experiments with ZINC, and the photovoltaics dataset (which by the way does not contain “very complex” molecules, [lines 225/226], but in fact simpler ones than ZINC, please rectify this!) are provided, which are better. The goal standard however is ChEMBL, and it would be highly recommended to run experiments in that dataset to replace QM9 if that’s still possible. Baselines. First, it is great that the authors compare against several VAE and other baselines. However, a simple autoregressive character-based SMILES-RNN baseline is missing. Even though this rather primitive model is less fancy than VAEs, in practice it provides not just syntactically, but also semantically correct molecules, which is in contrast to those by e.g. GrammarVAE or SMILES-VAE, which tend to generate semantically problematic molecules, which can likely never be made in the lab. Already the Li et al paper indicated (even though hidden in the Appendix!), that the KL divergence of the property distributions of generated vs. test molecules, for several key properties, is actually lower for the SMILES-RNN than for their autoregressive graph model. Also, in another recent paper ( https://doi.org/10.1186/s13321-018-0287-6 ), it turned out the SMILES-based RNNs are actually competitive with graph-based conditional models. Therefore, in my opinion, to claim state of the art, it is absolutely necessary (and straightforward, because trivial to implement, Also note that the SMILES-RNN does not need any masking, or sophisticated preprocessing!), to compare to a well-tuned SMILES-based LSTM (start with 3 layers x 512 dim) – also keeping in mind the results by Melis et al. ICLR 2018 for language models. In practice, the SMILES-RNN is widely and successfully used, and actually molecules designed by it have been made in the lab and tested (https://doi.org/10.1002/minf.201700153) Furthermore, in addition to Atom, Bond and Ring Countx, which are not very meaningful on their own, please add histograms for the following other physicochemical descriptors: # requires rdkit # from rdkit.Chem import Descriptors descriptors = [Descriptors.MolWt, Descriptors.BertzCT, Descriptors.NumHDonors, Descriptors.NumHAcceptors, Descriptors.MolLogP, Descriptors.TPSA, Descriptors.NumRotatableBonds] for descriptor in descriptors: value = descriptor(mol) # now collect value ... ## Style and Layout It would be desirable to have larger, proper histogram plots of the property comparison.